# Association between Advanced Lung Inflammation Index and 30-Day Mortality in Patients with Acute Respiratory Distress Syndrome

**DOI:** 10.3390/medicina57080800

**Published:** 2021-08-04

**Authors:** Manbong Heo, Jonghwan Jeong, Ire Heo, Sunmi Ju, Seungjun Lee, Yiyeong Jeong, Jongdeog Lee, Hocheol Kim, Jungwan Yoo

**Affiliations:** 1Division of Pulmonology and Allergy, Department of Internal Medicine, Gyeongsang National University Hospital, Gyeongsang National University School of Medicine, Jinju 52727, Korea; cough@kakao.com (M.H.); bechem6939@naver.com (J.J.); smangel518@naver.com (S.J.); juny2278@naver.com (S.L.); dr202202@naver.com (Y.J.); ljd8611@nate.com (J.L.); 2Division of Pulmonology and Allergy, Department of Internal Medicine, Gyeongsang National University School of Medicine and Gyeongsang National University Changwon Hospital, Changwon 51472, Korea; h2hawk@naver.com (I.H.); hochkim@gnu.ac.kr (H.K.)

**Keywords:** advanced lung inflammation index, acute respiratory distress syndrome, mortality

## Abstract

*Background and objectives:* The advanced lung inflammation index (ALI) was developed to assess the degree of systemic inflammation and has an association with prognosis in patients with lung malignancy. The prognostic value of ALI has not yet been evaluated in patients with acute respiratory distress syndrome (ARDS). *Materials and Methods:* Between January 2014 and May 2018, patients with ARDS in the medical intensive care unit (ICU) were reviewed retrospectively. The ALI value was calculated as the (body mass index × serum albumin level)/neutrophil-lymphocyte ratio. The cut-off value for distinguishing low from high ALI was defined according to receiver-operating characteristic curve analysis. *Results:* A total of 164 patients were analyzed. Their median age was 73 years, and 73% was male. The main cause of ARDS was pneumonia (95.7%, 157/164). ICU and in-hospital mortality rates were 59.8% (98/164) and 64% (105/164), respectively. The 30 day mortality was 60.9% (100/164). The median ALI value in non-survivors was lower than that in survivors at 30 day (3.81 vs. 7.39, *p* = 0.005). In multivariate analysis, low ALI value (≤5.38) was associated with increased 30 day mortality (odds ratio, 2.944, confidence interval 1.178–7.355, *p* = 0.021). *Conclusions:* A low ALI value was associated with increased 30 day mortality in patients with ARDS.

## 1. Introduction

Acute respiratory distress syndrome (ARDS) is characterized by non-cardiogenic pulmonary edema induced by inflammation, resulting in serious respiratory complication [1]. ARDS is commonly encountered in the intensive care unit (ICU). A recent study has reported that 1 out of 10 patients admitted to the ICU, and about one fourth of patients receiving mechanical ventilation had ARDS. Moreover, patients with ARDS have a high morbidity and mortality [2,3,4,5,6].

Systemic or pulmonary inflammation plays a major role in the development and progression of ARDS [7,8] and also affects clinical outcomes. It is vital to identify new or repurposed inflammatory markers that can reflect outcomes of ARDS.

The advanced lung inflammation index (ALI), which is based on body mass index (BMI), serum albumin and the neutrophil-lymphocyte ratio (NLR), is a known prognostic marker for patients with lung malignancy [9,10]. Each component of the ALI is also individually associated with the clinical outcomes of ARDS [11,12,13,14], but the clinical relevance of the ALI on mortality in patients with ARDS has not yet been elucidated. The aim of this study was to determine whether the ALI was associated with 30 day mortality in them.

## 2. Materials and Methods

### 2.1. Patients

From January 2014 through June 2018, medical records of patients with ARDS who had invasive mechanical ventilation in a medical intensive care unit (MICU) at a university-affiliated hospital were retrospectively reviewed. All these patients suffered from ARDS, requiring immediate invasive mechanical ventilation as supportive care. They received invasive mechanical ventilation at least ≥24 h. The Berlin definition criteria was fulfilled for diagnosis of ARDS [15]. Baseline (age, gender, comorbidities, body mass index (BMI)) and clinical (APACHE (acute physiology and chronic health evaluation) II score, SOFA (sequential organ failure assessment) score, septic shock, treatment, etc.) characteristics as well as laboratory parameters (neutrophil and lymphocyte count, hemoglobin concentration, platelet count, C-reactive protein level, total protein and albumin level, arterial blood gas analyses, etc.) and 30 day mortality were analyzed. ALI was calculated at the time of admission to the MICU and application of invasive mechanical ventilation. ALI was defined using the following formula: BMI × albumin/NLR.

This study was approved by the Institutional Review Board of Gyeongsang National University Hospital (approval GNUH2019-11-014). Because this study was conducted retrospectively, informed consent was waived. This study was performed following the ethical standards of institutional and/or national research committees and the Helsinki Declaration and its later amendments or comparable ethical standards.

### 2.2. Statistical Analysis

Categorical data were presented as median with interquartile range (IQR), and the Mann–Whitney U test was used to compare them. Non-categorical data were expressed as numbers with percentages and analyzed by using the chi-square or Fisher’s exact method. The 30 day mortality was compared between low and high ALI groups using the Kaplan–Meier method and log-rank test.

The cut-off value was determined to distinguish between low and high ALI according to its sensitivity and specificity by using a receiver-operating characteristic (ROC) curve and the Youden method [16]. Multivariate logistic regression analysis was used to evaluate factors associated with the 30 day mortality in patients with ARDS. Multivariate logistic regression analysis was performed entering variables with a *p*-value of less than 0.1 on univariate analyses. Differences with *p* < 0.05 were considered statistically significant. All data were analyzed using the SPSS software version 18.0 (SPSS Inc, Chicago, IL, USA).

## 3. Results

Figure 1 present flow diagram for patients’ inclusion and exclusion. During study period, 850 patients were admitted to MICU for mechanical ventilation support. Among them, 164 patients diagnosed with ARDS receiving invasive mechanical ventilation were included in this study.

The median age of patients was 73 years, and 73.2% were men. In total, 100 of the 164 patients (60.9%) had died at 30 days. Table 1 compares baseline and clinical characteristics between survivors and non-survivors at 30 days. Non-survivors were older and higher APACHE II and SOFA scores than those of survivors. Severe ARDS, septic shock and acute kidney injury were more frequently observed in non-survivors. Laboratory and ventilator parameters at initiation of invasive mechanical ventilation are shown and compared in Table 2. At 30 days, survivors had significantly higher hemoglobin concentration, ALI values, albumin levels and PaO_2_/FiO_2_ ratios than non-survivors. The NLR value and PaCO_2_ and FiO_2_ levels were lower in survivors than in non-survivors. Figure 2 shows a comparison of ALI values between survivors and non-survivors. In clinical parameters, there was significantly higher ALI in patients with shock than those without shock (4 [2.2–12.2] vs. 6.9 [3.9–13.9], *p* = 0.017).

ROC curve analysis was performed to select the cut-off value for classifying patients into high- or low-ALI groups. A cut-off value of 5.38 (specificity: 0.625, sensitivity: 0.680) was identified to discriminate 30 day mortality, using the area under the ROC curve (AUC: 0.629; 95% CI: 0.550–0.703, *p* = 0.0046). Among the 164 patients, 92 (56.1%) patients were designated as having low ALI, and 72 (43.9%) as having high ALI. The 30 day mortality rate was greater in the low ALI than the high ALI group (73.9% (68/92) vs. 44.5% (32/72), *p* < 0.001).

Factors associated with 30 day mortality were analyzed in univariate and multivariate method and are shown in Table 3. The APACHE II score (odds ratio (OR) 1.184, 95% confidence interval (CI) 1.077–1.303, *p* = 0.001, hemoglobin (OR 0.802, 95% CI 0.656–0.981, *p* = 0.032), partial pressure of carbon dioxide (OR 1.086, 95% CI 1.301–1.143, *p* = 0.002), and low ALI value (OR 3.001, 95% CI 1.1–7.355, *p* = 0.019), were significantly associated with increased 30 day mortality in patients with ARDS. The Kaplan–Meier survival curve showed that the low ALI group had a lower survival rate than the high ALI group (Figure 3, *p* < 0.001).

## 4. Discussion

This study showed that the median ALI value was lower in non-survivors than in survivors at 30 days, significantly. Furthermore, a low ALI value was correlated with a higher 30 day mortality.

Previous studies have reported that inflammation play an impact on the prognosis of various diseases, including cardiovascular diseases, malignancies, etc. [17,18]. Inflammation is also the crucial mechanism which ARDS develops and progresses [19]. Early recognition and alleviation of inflammation is important to manage patients with ARDS. Therefore, identification of new inflammatory markers of ARDS is essential.

Neutrophils are involved in the inflammatory process in ARDS [20,21]. Therefore, development or repurposing of a neutrophil-related index may hold promise as a prognostic marker in ARDS. The NLR, which reflects systemic inflammation, is associated with various diseases [22,23,24,25]. Recent studies have reported that the NLR is associated with mortality in ARDS [11]. BMI is a commonly used demographic characteristics that can be calculated easily if the patient’s height and weight are measured. BMI has been used as a screening index for overweight or obesity [26]. In obese individuals, adipose tissue produces various pro-inflammatory cytokines, including tumor necrosis factor-alpha, interleukin-1 beta, IL-6 and IL-8 [27]. High BMI may indicate a persistent inflammatory state [28]. BMI is also associated with outcomes in critically ill patients [29,30,31]. A recent meta-analysis revealed that high BMI was associated with decreased mortality in patients with ARDS [13]. Serum albumin, a value measured in hospitalized patients, has a role as acute phase reactants [32] and has been proposed as a reliable predictor of outcomes in critically ill patients with infectious diseases [33,34]. Hypoalbuminemia contributes to increased permeability edema in ARDS [35]. A previous study has shown that low albumin levels were related to the development and progression of ARDS [14].

The ALI, a formula combining the above three factors, was initially developed for evaluating systemic inflammation in lung cancer patients, and several studies have shown that the ALI was associated with outcomes in lung malignancy [9,10]. Each parameter that forms part of the ALI is individually associated with outcomes in patients with ARDS, but there have been no studies that evaluated the association between the ALI and ARDS. The utility of ALI in ARDS had not been evaluated previously. The current study revealed that the ALI score was lower in non-survivors with ARDS at 30 days, and that a low ALI was associated with 30 day mortality in multivariate analysis. The low ALI group had a lower survival rate than the high ALI group.

Some limitations existed in this study. First, because this study was performed retrospectively in a single center, selection bias could not be excluded and small cases of ARDS also limits a concrete conclusion for the utility of ALI, which make ALI difficult to generalize the findings to all patients with ARDS. Second, the mortality of patients with ARDS in this retrospective cohort was high. This might be due not only to the high severity of illness at mechanical ventilation but also to the low use of a rescue therapy, such as extracorporeal membrane oxygenation (ECMO). Considering the high cost and questioning survival benefit of ECMO, some patients’ family or relatives may refuse the use of ECMO, despite it being indicated as a rescue therapy, which may affect the high mortality rate in patients with ARDS. Third, the ALI score at MICU admission was calculated at one time point only, and a sequential ALI score was not measured. Whether changes in ALI scores have an impact on clinical outcomes in patients with ARDS needs to be further evaluated. Fourth, the important ventilator parameters such as lung compliance, plateau pressure or driving pressure were not recorded and monitored, and the association between ALI value and respiratory parameters was not evaluated in this study.

## 5. Conclusions

The ALI value was lower in non-survivors than in survivors at 30 days, and a low ALI value was associated with 30 day mortality in patients with ARDS. This study suggests that the low ALI may be a possible marker to be associated with prognosis in patients with ARDS.

## Figures and Tables

**Figure 1 medicina-57-00800-f001:**
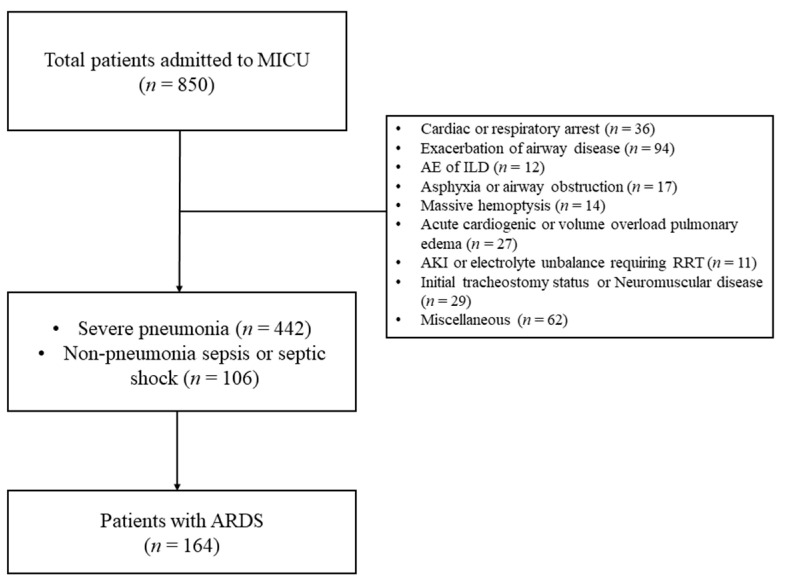
Flow diagram for patients’ inclusion and exclusion. (MICU, medical intensive care unit; ARDS, acute respiratory distress syndrome; AE, acute ex acerbation; ILD, interstitial lung disease; AKI, acute kidney injury; and RRT, renal replacement therapy).

**Figure 2 medicina-57-00800-f002:**
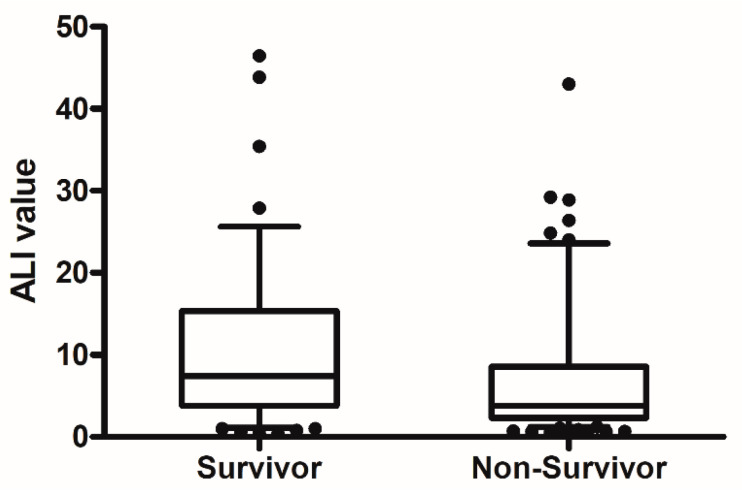
The comparison of ALI value between non-survivors and survivors at 30 day. (*p* = 0.005).

**Figure 3 medicina-57-00800-f003:**
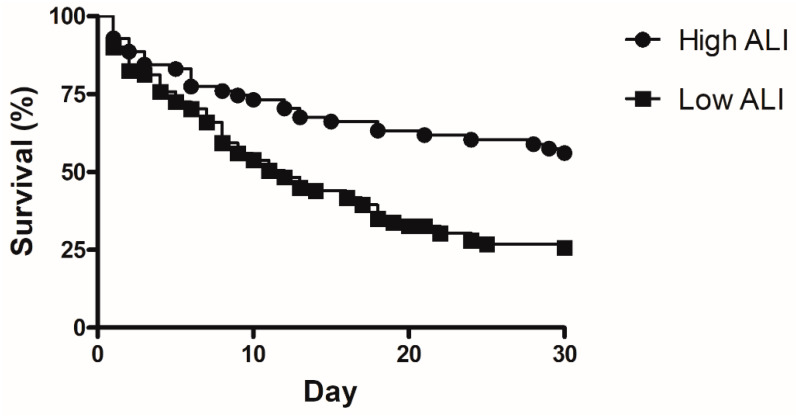
Kaplan–Meier curve for comparison of 30 day mortality between low ALI and high ALI. (*p* < 0.001).

**Table 1 medicina-57-00800-t001:** Baseline characteristics of total patients, survivor and non-survivors at 30 days.

Variables	Total	Survivor	Non-Survivors	*p*-Value
	N = 164	N = 64	N = 100	
Age, years old	73 (60–79.8)	66 (53.8–76.8)	75 (65.3–80)	0.001
Gender, male	120 (73.2)	48 (75)	72 (72)	0.67
Body mass index (kg/m^2^)	21.5 (19–24.6)	22.3 (20.4–25.2)	20.8 (18.6–24)	0.019
Diabetes mellitus	47 (28.7)	19 (29.7)	28 (28)	0.82
Ischemic heart disease	12 (7.3)	4 (6.3)	8 (8)	0.77
Heart failure	5 (3)	2 (3.1)	3 (3.0)	1
Chronic kidney disease	11 (6.7)	4 (6.3)	7 (7)	1
Chronic liver disease	21 (12.8)	10 (15.6)	11 (11)	0.39
Cerebrovascular disease	29 (17.7)	15 (23.4)	14 (14)	0.12
Active malignancy	18 (11)	7 (10.9)	11 (11)	0.99
COPD	16 (9.8)	7 (10.9)	9 (9)	0.68
APACHE II	26 (22–26)	23 (20–26)	30 (25–34)	<0.001
SOFA	12 (10–14)	11 (9–13)	13 (11–15)	<0.001
ARDS severity				0.001
mild	16 (9.8)	12(18.8)	4 (4)	
moderate	88 (53.7)	37 (57.8)	51 (51)	
severe	60 (36.6)	15 (23.4)	45 (45)	
Septic shock	118 (72)	39 (60.9)	79 (79)	0.012
Cardiac arrest	14 (8.5)	6 (9.4)	8 (8)	0.76
Acute kidney injury	106 (64.6)	31 (48.4)	75 (75)	0.001
RRT	48 (29.3)	9 (14.1)	39 (39)	0.32
Transfusion of blood products *	37 (22.6)	10 (15.6)	27 (27)	0.089
Steroid	37 (22.6)	11 (17.2)	26 (26)	0.19
Prone position	6 (3.7)	2 (3.1)	4 (4)	1
ECMO	16 (9.8)	7 (10.9)	9 (9)	0.68

BMI, body mass index; COPD, chronic obstructive pulmonary disease; APACHE, acute physiology and chronic health evaluation; SOFA, sequential organ failure assessment; ARDS, acute respiratory distress syndrome; AKI, acute kidney injury; RRT, renal replacement therapy; and ECMO, extracorporeal membrane oxygenation. * Blood products includes red blood cells, fresh frozen plasma or platelets.

**Table 2 medicina-57-00800-t002:** Comparisons of laboratory and ventilator values at intubation and mechanical ventilation between survivors and non-survivors at 30 days.

Variables	Total	Survivors	Non-Survivors	*p*-Value
	N = 164	N = 64	N = 100	
NLR	11.8 (5.6–20.9)	9.2 (4.8–17.9)	14.2 (7.2–21.8)	0.048
Hb, g/dL	11.5 (9.6–12.8)	12.2 (10.4–13.5)	11 (9.4–12.6)	0.01
Platelet, ×10^3^/mm^3^	184 (127.2–290.5)	194.5 (141.5–275.8)	162 (110.2–291)	0.35
ALI	4.51 (2.53–12.32)	7.39 (3.79–15.36)	3.81 (2.29–8.56)	0.005
Albumin, g/dL	2.6 (2.3–3.1)	2.8 (2.5–3.3)	2.6 (2.2–2.9)	0.001
CRP, mg/dL	17.9 (9.5–27.1)	18.6 (11.3–27.6)	17.7 (8.9–26.4)	0.44
Lactate, mmol/L	3.2 (2–5.8)	2.8 (1.7–5.5)	3.5 (2.1–6.1)	0.124
pH	7.32 (7.22–7.4)	7.36 (7.29–7.42)	7.28 (7.19–7.39)	<0.001
PaCO_2_, mmHg	38 (32–45)	35 (30–41.7)	41 (33.2–46)	0.002
Bicarbonate, mmol/L	18 (16–22)	19.5 (17–21)	18 (15–22)	0.228
P/F ratio	115 (84.1–153.8)	130 (100–194.8)	106 (77–142.5)	0.002
Vt, mL	6.85 (6.02–7.89)	7.31 (6.25–8.27)	6.75 (6.01–7.79)	0.077
PEEP, cmH_2_O	8 (5–10)	8 (5–10)	8.5 (6–10)	0.21
FiO_2_	0.8 (1–0.6)	0.6 (0.52–0.83)	0.8 (0.6–1)	0.002

NLR, neutrophil/lymphocyte ratio; Hb, hemoglobin; ALI, advanced lung inflammation index; CRP, C-reactive protein; NT-proBNP, N terminal probrain natriuretic peptide; PaCO_2_, partial pressure of carbon dioxide; PF, partial pressure of oxygen/fractioned inspired oxygen; Vt, tidal volume; PEEP, positive end-expiratory pressure; and FiO_2_, fractioned inspired oxygen.

**Table 3 medicina-57-00800-t003:** Univariate and multivariate analysis for factor associated with 30 day mortality.

	Univariate			Multivariate		
Variable	OR	95% CI	*p*-Value	OR	95% CI	*p*-Value
Age	1.034	1.010–1.058	0.004	1.014	0.984–1.044	0.364
APACHEII	1.219	1.134–1.310	<0.001	1.184	1.077–1.303	0.001
Shock	2.411	1.203–4.834	0.013	1.364	0.529–3.514	0.520
AKI	3.194	1.639–6.224	0.001	2.001	0.748–5.351	0.167
Hb, g/dL	0.825	0.713–0.954	0.009	0.802	0.656–0.981	0.032
PaCO_2_, mmHg	1.055	1.018–1.093	0.003	1.086	1.031–1.143	0.002
P/F ratio	0.989	0.983–0.995	0.001	0.997	0.988–1.005	0.424
Low ALI value	3.542	1.835–6.837	<0.001	3.001	1.202–7.492	0.019

The multivariate model showed a good calibration as assessed by Hosmer–Lemeshow goodness of fit test, chi-square = 1.515, *p* = 0.992. OR, odds ratio; CI, confidence interval; APACHE, acute physiology and chronic health evaluation; AKI, acute kidney injury; Hb, hemoglobin; PaCO_2_, arterial partial pressure of carbon dioxide; P/F, arterial partial pressure of oxygen/fractional inspired oxygen; ALI, advanced lung inflammation index; and Hb, hemoglobin.

## Data Availability

The data that support the findings of this study are available from the corresponding author, J.Y., upon reasonable request.

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
