# Peer review of "Association between Advanced Lung Inflammation Index and 30-Day Mortality in Patients with Acute Respiratory Distress Syndrome"

_medicina, 2021, doi:10.3390/medicina57080800_

Round 1
Reviewer 1 Report
I'm happy with authors reply
Reviewer 2 Report
I appreciate their sincere efforts made by the authors to respond to my concerns and suggestions, which improve their work.
Authors have addressed my questions and concerns adequately during the revision.
This manuscript is a resubmission of an earlier submission. The following is a list of the peer review reports and author responses from that submission.
Round 1
Reviewer 1 Report
I like to thank you for having the possibility reviewing this paper; I have several comments:
- What is the real clinical imapct of the paper? This has to be stated and discussed!
- What are real new information? This has to be stated and discussed!
- The paper has serious limitations - more than just the retrospective design! These limitations have to be mentioned and discussed in a more profound way.
Author Response
Thank you for your review of our manuscript. We try to respond to your comments by a point-by-point
Point 1: What is the real clinical impact of the paper? This has to be stated and discussed!
Response 1: Thank you for your valuable comments. Advanced lung cancer inflammation (ALI) originally developed to assess degree of systemic inflammation at the time of diagnosis in metastatic non-small cell lung (NSCLC) cancer patients. ALI was associated with clinical outcomes in patients with lung cancer. (BMC Cancer 2013; 13: 158, Clin Respir J 2018; 12: 2013-2019). Systemic inflammation is also important mechanism regarding development and progression of ARDS. Components (BMI, blood neutrophil, lymphocyte counts and albumin) of index is not required based on experimental method but are feasible parameters to measure in clinical practice This index has not been tested in patients with ARDS where inflammation is also important mechanism regarding development and progression of ARDS. We wonder whether ALI, simple inflammation index, has clinical utility to evaluate 30-day mortality in ARDS.
Point 2: What are real new information? This has to be stated and discussed!
Response 2: Thank you for your valuable comments.
ALI is easily calculated in clinical practice. We found that the clinical utility of ALI is not limited to lung cancer, but expanded to ARDS. ALI is likely to be a potential and repurposed index to predict outcome in patients with ARDS. We think this is a new information or point.
Point 3: The paper has serious limitations - more than just the retrospective design! These limitations have to be mentioned and discussed in a more profound way.
Response 3: Thank you for your valuable comment. We added sentences for limitation as you advised like this.
Page 7, line 29-30 “Small cases of ARDS also limits a concret conclusion for utility of ALI.
Page 7, line 40-41 “Fourth, the important ventilator parameters such as lung compliance, plateau pressure or driving pressure were not recorded and monitored.”
Reviewer 2 Report
In this manuscript authors evaluate the use of the advanced lung inflammation index (ALI), extended in lung cancer, to further explore its utility as mortality predictor in ARDS patients. In this retrospective study, authors concluded that low ALI values are correlated to higher 30-days mortality. The paper is well written and results are interesting for the clinical practice. I have no further request.
Author Response
Thank you for your comment and we appreciate it
Reviewer 3 Report
This study by Heo et al. was designed to investigate if advanced lung inflammation index (ALI) may be associated with increased 30-day mortality in patients with ARDS as marker that can predict the clinical outcome. However, there are some concerns that should be appropriately addressed.
Major comments:
- Authors point that they included in the study 164 patients diagnosed with ARDS. However, what was the number of patients enrolled in the study before to select only these 164? What were the criteria used to exclude patients? Inclusion of a flow diagram for patient inclusion and exclusion where authors present the patients distribution from the database used could help to the reader to understand the different steps followed in the design and selection of the patients cohort.
- Authors point that “ Patients with ARDS who received invasive mechanical ventilation…. were retrospectively reviewed”. However, they do not describe if patients developed ARDS after mechanical ventilation or if these patients suffered ARDS previously and they needed mechanical ventilation as supportive care. In addition, patients included in the study were mechanical ventilated for similar time? It is an important issue, because ventilation with high airway pressure and tidal volume is an important risk factor for respiratory failure. Therefore, authors should include information about the mechanical ventilation time or if they selected patients with any specific range of ventilation such as ≥ 48 hours.
- There are other risk factor associated with ARDS independent of mechanical ventilation, such as arterial pH, bicarbonate and lactate in addition to PaCO2 as indicators of gas exchange and metabolism as well as transfusion of blood products such as red blood cells, plasma or platelets. Therefore, Inclusion of these parameters in their study will reinforce the association with ALI.
- The authors have compared ALI values between non-survivors and survivors at 30-days as they showed in the Table 1 and Figure 1. However, the do not have performed the association of ALI with the clinical parameters. This comparison will reinforce the use of ALI as prognostic value.
Minor:
Table 2 and 3, some units are missed. Authors should include the units of the parameters measured such as Hb, Platelet, Albumin.
Author Response
Thank you for your valuable comments for our manuscript. We appreciated your interest of our manuscript and having a chance to revise it once again.
Point 1: Authors point that they included in the study 164 patients diagnosed with ARDS. However, what was the number of patients enrolled in the study before to select only these 164? What were the criteria used to exclude patients? Inclusion of a flow diagram for patient inclusion and exclusion where authors present the patients distribution from the database used could help to the reader to understand the different steps followed in the design and selection of the patients cohort.
Response 1: Thank you for your valuable comment. We added flow diagram (new figure 1) for patients inclusion and exclusion as you advised.
To clarify, We added sentences and made a new figure 1
Page 3, line 3-13 “Figure 1 present flow diagram for patients’ inclusion and exclusion. During study period, 850 patients were admitted to MICU for mechanical ventilation support. Among them, 164 patients diagnosed with ARDS receiving invasive mechanical ventilation were included in this study.”
Point 2: Authors point that “Patients with ARDS who received invasive mechanical ventilation…. were retrospectively reviewed”. However, they do not describe if patients developed ARDS after mechanical ventilation or if these patients suffered ARDS previously and they needed mechanical ventilation as supportive care. In addition, patients included in the study were mechanical ventilated for similar time? It is an important issue, because ventilation with high airway pressure and tidal volume is an important risk factor for respiratory failure. Therefore, authors should include information about the mechanical ventilation time or if they selected patients with any specific range of ventilation such as ≥ 48 hours.
Response 2: Thank you for your valuable comment. Patient suffered from acute respiratory failure accompanied by bilateral pulmonary infiltrates in their CXR or chest CT scan, not fully explained by volume overload or cardiogenic pulmonary edema before invasive mechanical ventilation. They were intubated and received invasive mechanical ventilation under at least 5cmH2O of PEEP. All patients received ventilation more than 24 hours after diagnosis of ARDS.
To clarify this, we added sentences in method as you advised
Page 2, line 11-13 “All patient these patients suffered ARDS requiring immediate invasive mechanical ventilation as supportive care. They received invasive mechanical ventilation at least ≥ 24 hours.”
Point 3: There are other risk factor associated with ARDS independent of mechanical ventilation, such as arterial pH, bicarbonate and lactate in addition to PaCO2 as indicators of gas exchange and metabolism as well as transfusion of blood products such as red blood cells, plasma or platelets. Therefore, Inclusion of these parameters in their study will reinforce the association with ALI.
Response 3: Thank you for your valuable comment. We added arterial pH, bicarbonate and lactate (Table2) as well as transfusion of blood products such as red blood cells, plasma or platelets (Table 1).
Point 4: The authors have compared ALI values between non-survivors and survivors at 30-days as they showed in the Table 1 and Figure 1. However, the do not have performed the association of ALI with the clinical parameters. This comparison will reinforce the use of ALI as prognostic value.
Response 4: Thank you for your valuable comment. We performed the association of ALI with the clinical parameters such as baseline characteristics and clinical presentation. Among baseline characteristics and clinical presentations, we found that patients with shock had significantly lower ALI than those without shock (4[ 2.2-12.2] vs 6.9 [3.9-13.9], P=0.017)
To clarify, we added sentence like this
Page 3, line 24-26 “In clinical parameters, there was significantly higher ALI in patients with shock than those without shock (4 [2.2-12.2] vs 6.9 [3.9-13.9], P=0.017)”
Point 5: Table 2 and 3, some units are missed. Authors should include the units of the parameters measured such as Hb, Platelet, Albumin.
Response 5: Thank you for your comment. We added units of parameters in Table 2 and 3 as you advised.